# Fundus autofluorescence features specific for *EYS*-associated retinitis pigmentosa

Taro Kominami [1☉*], Tien-En Tan [2☉], Hiroaki Ushida[1], Kanika Jain[3], Kensuke Goto[1], Yasmin M. Bylstra[4], Ai Fujita Sajiki [1,5], Ranjana S. Mathur[2], Junya Ota [1], Weng Khong Lim[4], Koji M Nishiguchi[1], Beau J. Fenner [2*]

**1** Department of Ophthalmology, Nagoya University Graduate School of Medicine, Nagoya, Aichi, Japan, **2** Singapore National Eye Centre, Singapore Eye Research Institute, and Ophthalmology and Visual Sciences Academic Clinical Program, Duke-NUS Graduate Medical School, Singapore, Singapore, **3** Genome Institute of Singapore, Singapore, Singapore, **4** Institute for Precision Medicine, SingHealth Duke-NUS Graduate Medical School, Singapore, Singapore, **5** Division for Advanced Medical Research, Center for Research of Laboratory Animals and Medical Research Engineering, Nagoya University Graduate School of Medicine, Nagoya, Aichi, Japan

☉ These authors contributed equally to this work as first authors.
* taro.kominami@gmail.com (TK); beaufenner@duke-nus.edu.sg (BJF)

## Abstract

### Purpose

To assess the utility of fundus autofluorescence (FAF) patterns for predicting the *EYS* genotype in retinitis pigmentosa (RP) patients.

### Methods

This retrospective, multi-institutional study analyzed FAF images from 200 RP patients (74 with *EYS* and 126 without *EYS*) from Singapore and Japan. Seven FAF patterns including the infinity sign and a broad banded hyper-autofluorescent leading edge were evaluated for their association with the *EYS* genotype.

### Results

The infinity sign and broad banded hyperautofluorescent leading edge occurred more frequently in *EYS* eyes (p = 0.0014 and p = 0.036 respectively). Logistic regression analysis showed that the infinity sign was predictive of *EYS* (p = 0.003). The combined FAF parameters predicted *EYS* with a specificity of 95.20%, sensitivity of 25.68% and accuracy of 69.50%, with a cut-off value 0.5 based on the probability of seven FAF parameters.

### Conclusions

In this multinational cohort study of patients with RP, we demonstrated that specific FAF patterns, particularly the infinity sign, have clinical utility in identifying patients with *EYS*-associated disease. These findings may be useful for clinicians and geneticists when genotyping patients with RP, and may also enhance our understanding of

**Data availability statement:** All relevant data for this study are publicly available from the Nagoya University Institutional Repository (http://hdl.handle.net/2237/0002011889).

**Funding:** All the funding or resources of support for this study were the Japan Society for the Promotion of Science (JSPS) KAKENHI, grant number 23K15929 to TK; the SingHealth Foundation grant number R1748/71/2020 to BJF; and the SingHealth - Duke-NUS Institute of Precision Medicine grant numbers 05/FY2020/EX/06-A41 and 05/FY2022/EX(SLP)/69-A131(b) to KJ, YMB, and WKL. Funding organizations had no role in the design or conduct of this study. There was no additional external funding received for this study.

**Competing interests:** The authors have declared that no competing interests exist.

underlying pathophysiology of *EYS*-associated RP, which is a prevalent cause of RP in Asia and elsewhere.

## Introduction

Retinitis pigmentosa (RP) is an inherited retinal dystrophy affecting approximately 1 in 4000 individuals globally [1–3]. RP typically presents with nyctalopia and peripheral vision loss secondary to rod photoreceptor degeneration, followed by loss of central vision secondary to cone photoreceptor loss [1]. Although the pathogenic genes of RP are highly diverse, pathogenic variants in the eyes shut homolog (*EYS*) gene are notably prevalent among East Asian populations, establishing *EYS* as a significant factor in RP's genetic landscape [4–9].

Fundus autofluorescence (FAF) imaging, particularly in ultra-widefield (UWF) format, has been shown to be useful in the diagnosis and monitoring of RP [10] and in evaluating structure-function correlation [11,12]. FAF features such as parafoveal rings of high-density autofluorescence in patients with RP [13] and the genotype-phenotype correlation [14] were previously reported. There is a study to report that some FAF features are detected in patients with *EYS*-associated RP [15], but specific FAF features correlating with pathogenic *EYS* variants are not fully defined. More focused research is necessary especially to understand the FAF characteristics unique to *EYS*-associated RP, which could significantly impact diagnostic strategies and management in East Asian RP populations [4–6,8,9].

In this study, we aimed to evaluate the utility of fundus FAF features in predicting the presence of pathogenic *EYS* variants. By analyzing a comprehensive dataset from multiple institutions, we sought to enhance the specificity of FAF imaging for *EYS*-associated retinitis pigmentosa, potentially streamlining the pathway to appropriate genetic testing and therapeutic interventions. Our findings demonstrate the significant role of specific FAF features, particularly the so-called "infinity sign", for the identification of *EYS*.

## Methods

This was a retrospective, multi-institutional observational study to identify characteristic FAF features in patients with *EYS* gene variants associated with RP. We reviewed the medical charts of RP patients who visited the Department of Ophthalmology at Nagoya University Hospital or the Department of Medical Retina at the Singapore National Eye Centre between 2010 and 2023. Written informed consent was obtained for all patients. The protocol of this study was conformed to the tenets of the Declaration of Helsinki. The study received approval from the institutional review boards/ ethics committee of Nagoya University Hospital (study number 2020-0598) and Singapore National Eye Centre (study number R1748/71/2020). Patients with a history of ocular surgery, other retinal disease, or if the FAF images were of insufficient quality for analysis were excluded. Data were accessed on 15/May/2023 for Singapore data and on 29/Oct/2023 for Japan data. We had access to information that could identify individual participants during or after data collection but all patient identifiers were removed from the data set used for the study.

### Patient demographics

We analyzed 200 FAF images from 200 eyes of patients diagnosed with RP sampled from the study centers. The study included RP cases of early-stage, late-stage, and cases with possible

early cone dysfunction. The study cohort included 74 eyes with pathogenic EYS variants (EYS) and 126 eyes without pathogenic *EYS* variants (non-EYS). In Singapore, 24 eyes of EYS and 76 eyes of non-EYS were sequentially enrolled with a confirmed genetic diagnosis, while in Japan, patients were randomly selected, with 50 eyes in each category for balance. All participants were selected based on their genetic diagnosis, confirmed through comprehensive genomic testing. The proportion of *EYS-* and non-*EYS*-associated RP in the pooled cohort was 0.37 and 0.65, respectively, resulting in a Z-score of –5.2 and a study power of 99.9% for detecting differences between *EYS-* and non-*EYS* associated RP based on a two-proportion Z-test.

## Fundus autofluorescence parameters

FAF imaging was performed using an ultra-wide field imaging device (Optos P200Tx; Optos, Dunfermline, UK). The presence of seven FAF parameters were evaluated (Fig 1): hyper autofluorescent (AF) crescent (Fig 1A), foveal hyper AF focus (Fig 1B), infinity sign (Fig 1C) which is characterized by nasal peripapillary and macula sparing resembling the shape of an infinity(∞) symbol, irregular sectoral areas of peripheral sparing (Fig 1D), broad-banded hyper AF leading edge (Fig 1E), inferior sectoral RP (Fig 1F), and a parafoveal hyper AF ring (Robson ring; Fig 1G). Four examiners (TK, TET, HU, and BJF) assessed the presence of the seven parameters in 200 FAF images with genotype masking at the time of grading. If each parameter was judged to be present in the image, a score of 2 was given; if it was judged not to be present, a score of 0 was given; if it was unclear whether it was present or not, a score of 1 was given. Inter-examiner variability of each parameter scored by four examiners was evaluated by the Fleiss' kappa coefficient and parameters with non-negligible variability with Fleiss' kappa coefficient lower than 0.2 were excluded from the analysis. For parameters that were judged not to have much inter-examiner variability with Fleiss' kappa coefficient larger than 0.2, the scores of the four examiners were averaged, and a score of 1 or greater was judged to be positive, and a score less than 1 was judged to be negative.

## Clinical outcomes

The primary outcome measure was the likelihood of occurrence of each features, in isolation or combination with other features, in eyes with and without *EYS* pathogenic variants.

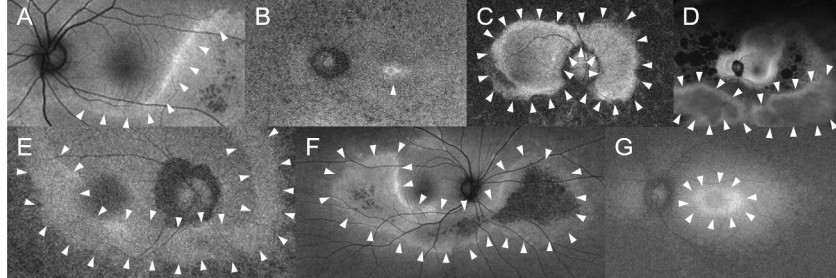

**Fig 1. Typical fundus autofluorescence images illustrating various diagnostic parameters.** This figure shows typical fundus autofluorescence (FAF) images for each of the diagnostic parameters used to evaluate *EYS*-associated retinitis pigmentosa. Each panel shows a unique feature characteristic of different retinal conditions: A: hyperautofluorescent crescent. B: focal foveal hyperautofluorescence. C: localized macular and peripapillary sparing (infinity sign). D: irregular sectoral areas of peripheral sparing. E: a broad banded hyper-autofluorescent leading edge. F: inferior sectoral retinitis pigmentosa. G: a parafoveal hyperautofluorescent ring (Robson ring). Arrowheads in each panel highlight the specific lesion areas.

Secondary outcomes included the sensitivity, specificity, and accuracy of these features for predicting EYS group, and were calculated by incorporating a categorical variables for each parameter into a logistic regression model for statistical analysis. Receiver operating curve (ROC) and area under the curve were calculated by logistic regression model.

## Statistical analysis

Inter-examiner variability was estimated using Fleiss kappa coefficients. Features with a kappa coefficient above 0.2 were included for further analysis. Differences in the presence of FAF features between EYS and non-EYS groups were assessed using Chi-square tests. Bonferroni correction was performed for multiple testing. Logistic regression was conducted to determine whether each feature, in isolation or in combination, was indicative of EYS or non-EYS groups. Sensitivity, specificity, and overall accuracy of the predictive model was calculated using a 2 x 2 contingency table, with a predefined cut-off value of 0.5. Python Scipy and Python Statsmodels were used for statistical analyses [16].

## Results

Our analysis involved 200 FAF images from 200 patients with RP, carrying either pathogenic *EYS* variants (EYS) or other genetic variants (non-EYS). The ratio of Singaporeans and Japanese differed between EYS group and non-EYS group, but there were no significant differences in age or gender between the two groups (Table 1). The EYS group had significantly better best corrected visual acuity (BCVA) at the time of retinal imaging compared to the non-EYS group. A list of causative genes in the non-EYS group is shown in Table 2. The top five causative genes associated with RP in the non-EYS group were: *USH2A*, *RP1*, *PRPH2*, *RHO* and *RPGR*. The most common pathogenic variants for *EYS* were c.4957dupA and c.2528G > A in the Japanese cohort, while the Singapore cohort was enriched for *EYS* c.6416G > A. The distributions of *EYS* variants are shown in S1 Fig.

### Inter-examiner variability and FAF feature reliability

Inter-examiner variability outcomes are shown in Table 3. Although there was some variation in the interquartile ranges, the medians were all 0. Irregular sectorial areas of peripheral sparing was the only parameter in which inter-examiner variability was not negligibly small and this feature was excluded from further analyses. Inter-examiner agreement for other FAF parameters indicated at least fair agreement (kappa ≥ 0.37), with consistent scoring among the four retina specialists who undertook the image evaluation.

### Comparative analysis of EYS and non-EYS groups for FAF parameters

The presence of localized macular and peripapillary sparing on FAF, forming a so-called "infinity sign", and a broad-banded hyper AF leading edge were over-represented in the EYS

**Table 1. Demographic characteristics of the patients (n = 74 EYS and n = 126 non-EYS).**

|  | EYS | non-EYS | p-value |
|---|---|---|---|
| SGP cases (%) | 24 (32.4) | 76 (60.3) | 0.00025 |
| Female gender (%) | 34 (45.9) | 58 (46.0) | 0.89 |
| Median age at imaging (IQR) | 51.5 (22.6) | 51.2 (26.5) | 0.92 |
| Median logMAR BCVA at imaging (IQR) | 0.13 (0.68) | 0.40 (0.85) | 0.001 |

SGP Singapore, IQR interquartile range, MAR minimum angle of resolution, BCVA best corrected visual acuity

group compared to the non-EYS group, with chi square tests yielding p values of p = 0.0014 and p = 0.036 after Bonferroni correction, respectively (Fig 2). Pathogenic genes of cases who were judged with "infinity sign" were 3 cases with *PRPH2*, one case with *RP1*, one case with *REEP6* and one case with *USH2A*, and those with a broad-banded hyper AF leading edges were 4 cases with *PRPH2*, one case with *RHO*, one case with *RP1L1* and one cases with *USH2A*. *EYS* variants detected in cases with an infinity sign and a broad-banded hyperAF leading edge are shown in the S1 Table.

**Table 2. Pathogenic genes of non-EYS group.**

| gene | Number of each gene (SGP/ JPN) |
|---|---|
| *USH2A* | 26 (22/ 4) |
| *RP1* | 16 (4/ 12) |
| *PRPH2* | 8 (5/ 3) |
| *RHO, RPGR* | 7 (3/ 4) |
| *PDE6B* | 5 (2/ 3) |
| *PRPF3* | 4 (4/ 0) |
| *PRPF31* | 4 (1/ 3) |
| *CLN7, REEP6, RP1L1* | 3 (3/ 0) |
| *CRX* | 3 (2/ 1) |
| *RP2* | 3 (0/ 3) |
| *ADGRV1, PRPF6* | 2 (2/ 0) |
| *TULP1* | 2 (1/ 1) |
| *ABCA4, RPE65* | 2 (0/ 2) |
| *ARHGEF18, C21ORF2, CEP290, CNGA1, HK1, IFT140, KLHL7, NRL, PCDH15, PDE6A, POMGNT1, PROM1, RLBP1, RP9, SNRNP200, SPATA7* | 1 (1/ 0) |
| *BBS2, IMPDH1, IMPG2, MAK, MERTK, RDH12, SAG, SEMA4A* | 1 (0/ 1) |

SGP Singapore, JPN Japan

**Table3. Median of each examiner's scoring and Fleiss' kappa coefficients of FAF image parameters.**

| | TK EYS (median, IQR) | TET EYS (median, IQR) | HU EYS (median, IQR) | BJF EYS (median, IQR) | TK nonEYS (median, IQR) | TET nonEYS (median, IQR) | HU nonEYS (median, IQR) | BJF nonEYS (median, IQR) | Fleiss' kappa coefficient | agreement |
|---|---|---|---|---|---|---|---|---|---|---|
| Hyper AF crescent | 0, 0 | 0, 1 | 0, 1 | 0, 0 | 0, 0 | 0, 0 | 0, 0 | 0, 0 | 0.65 | substantial |
| Foveal hyperAF focus | 0, 0 | 0, 1 | 0, 1 | 0, 0 | 0, 0 | 0, 1 | 0, 1 | 0, 0 | 0.53 | moderate |
| Infinity sign (nasal peripapillary sparing) | 0, 1 | 0, 1 | 0, 1 | 0, 1 | 0, 0 | 0, 0 | 0, 0 | 0, 0 | 0.51 | moderate |
| Inferior sectoral RP | 0, 0 | 0, 0 | 0, 0 | 0, 0 | 0, 0 | 0, 0 | 0, 0 | 0, 0 | 0.43 | moderate |
| Broad banded hyperAF leading edge | 0, 1 | 0, 1 | 0, 0 | 0, 0 | 0, 0 | 0, 0 | 0, 0 | 0, 0 | 0.37 | fair |
| Irregular sectoral areas of peripheral sparing | 0, 0 | 0, 0 | 0, 0 | 0, 0 | 0, 0 | 0, 0 | 0, 0 | 0, 0 | 0.08 | slight |
| Robson ring | 0, 0 | 0, 1 | 0, 0 | 0, 0 | 0, 0 | 0, 1 | 0, 0 | 0, 0 | 0.53 | moderate |

FAF fundus autofluorescence, AF autofluorescence, RP retinitis pigmentosa, IQR interquartile range

## Prediction of EYS-associated RP based on combinations of fundus autofluorescence features

The infinity sign exhibited the highest positive coefficient (beta 0.598, p = 0.003), indicating a strong predictive value for identifying *EYS*-associated RP (Table 4). Fig 3A shows the probability that an FAF image belonged to a patient with *EYS*-associated RP. This probability was calculated by substituting the presence or absence of each parameter into a logistic regression equation. A 2 × 2 contingency table with a cut-off value of 0.5 was then used to determine whether a case was EYS or non-EYS (Fig 3B). Sensitivity, specificity, and accuracy of the model was calculated as 25.68%, 95.24%, and 69.50%, respectively. The high specificity suggests that the model is effective at identifying true negatives, which is crucial for ruling out non-EYS RP in patients. The ROC curve created by combining FAF parameters was improved over the ROC curve created by using only visual acuity, age (at imaging and onset), and sex, with the AUC improving from 0.42 to 0.53 as shown in S2 Fig.

## Representative FAF images of patients with EYS-associated RP

Representative FAF images from patients with *EYS*-associated RP are shown in Fig 4. All FAF images were determined to have an infinity sign and broad banded hyper AF leading edge. A hyper-AF crescent was detected in the FAF images shown in Fig 4A, 4B and 4D.

A: Japanese patient, 67-year-old female with a best corrected visual acuity (BCVA) of 0.1 (logMAR). Genetic variants c.2528G > A/c.6557G > A were detected. Positive for "

| A Hyper AF crescent P = 0.026 | | prediction | |
|---|---|---|---|
| | | positive | negative |
| label | EYS | 12 | 62 |
| | non-EYS | 7 | 119 |

| B Foveal hyperAF focus P = 0.52 | | prediction | |
|---|---|---|---|
| | | positive | negative |
| label | EYS | 10 | 64 |
| | non-EYS | 12 | 114 |

| C Infinity sign (nasal peripapillary sparing) P = 0.00024 | | prediction | |
|---|---|---|---|
| | | positive | negative |
| label | EYS | 17 | 57 |
| | non-EYS | 6 | 120 |

| D Inferior sectoral RP P = 0.78 | | prediction | |
|---|---|---|---|
| | | positive | negative |
| label | EYS | 4 | 70 |
| | non-EYS | 7 | 119 |

| E Broad banded hyperAF leading edge P = 0.0060 | | prediction | |
|---|---|---|---|
| | | positive | negative |
| label | EYS | 15 | 59 |
| | non-EYS | 8 | 118 |

| F Robson ring P = 0.59 | | prediction | |
|---|---|---|---|
| | | positive | negative |
| label | EYS | 10 | 64 |
| | non-EYS | 22 | 104 |

**Fig 2. Evaluation of EYS predictive indicators using multiple 2 × 2 contingency tables.** This figure shows a series of 2 × 2 contingency tables analyzing the relationship between various predictive indicators and EYS across different imaging features. A: hyperautofluorescent crescent, B: focal foveal hyperautofluorescence, C: localized macular and peripapillary sparing (infinity sign), D: inferior sectoral retinitis pigmentosa, E: a broad banded hyper-autofluorescent leading edge, F: a parafoveal hyperautofluorescent ring (Robson ring). Each table categorizes the outcomes into positive and negative predictions for EYS, with counts of true positives, false positives, true negatives, and false negatives listed. The rows represent the presence or absence of *EYS* pathogenic variants, and columns categorize the prediction outcomes. P-values calculated by the chi-square test to evaluate the difference between the EYS and non-EYS groups are shown under each parameter name.

**Table 4. Results of logistic regression analysis.**

| parameters | coefficient | SE | z | P-value | 95% confidence interval |
|---|---|---|---|---|---|
| Hyper AF crescent | 0.3346 | 0.26 | 1.286 | 0.199 | [−0.176, 0.845] |
| Foveal hyperAF focus | 0.2246 | 0.164 | 1.373 | 0.17 | [−0.096, 0.545] |
| Infinity sign | 0.598 | 0.203 | 2.943 | 0.003 | [0.200, 0.996] |
| Broad banded hyperAF leading edge | 0.1522 | 0.194 | 0.785 | 0.432 | [−0.228, 0.532] |
| Robson ring | −0.1065 | 0.185 | -0.575 | 0.565 | [−0.470, 0.257] |
| Inferior sectoral RP | −0.1815 | 0.231 | -0.785 | 0.433 | [−0.635, 0.272] |

SE standard error, AF autofluorescence, RP retinitis pigmentosa

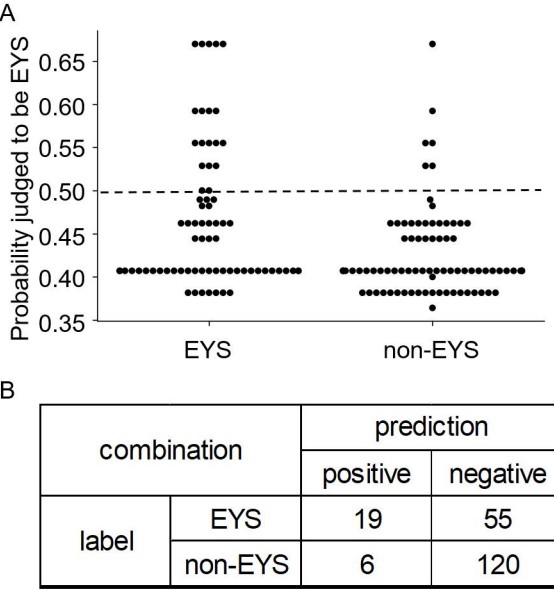

Fig 3. **Analysis of EYS prediction using logistic regression.** This figure illustrates the prediction results for EYS using logistic regression analysis. A displays shows swarm plots of probabilities calculated by the logistic regression equation, indicating the likelihood of EYS versus non-EYS. B shows a 2×2 contingency table where a cutoff value of 0.50 determines the prediction of EYS (0.50 or higher) versus non-EYS (less than 0.50). The table entries represent the counts of true positives (n = 19), true negatives (n = 120), false positives (n = 55), and false negatives (n = 6), indicating the performance of the predictive model.

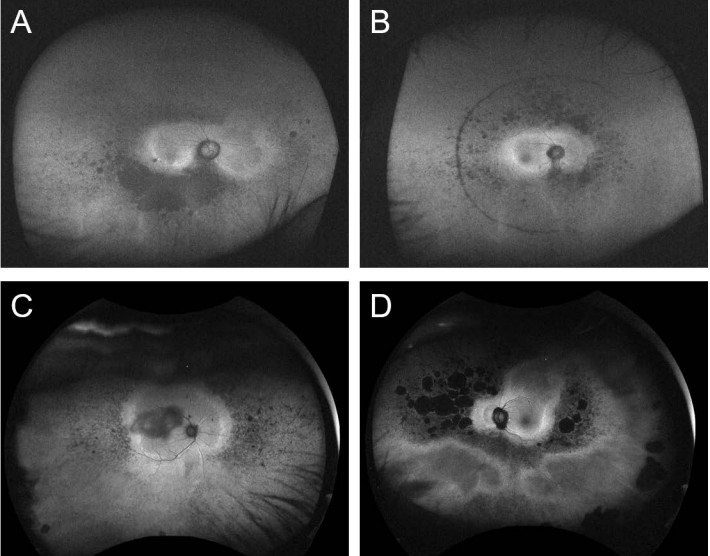

Fig 4. **Representative fundus autofluorescence images of EYS group.** This figure shows fundus autofluorescence images from different cases of individuals diagnosed with pathogenic *EYS* variants, highlighting the presence of specific autofluorescence patterns.

hyperautofluorescent crescent," "infinity sign," and "a broad banded hyper-autofluorescent leading edge." B: Japanese patient, 45-year-old male with a BCVA of 0.0 (logMAR). Genetic variants c.4957dupA/c.8805C > A were detected. Features include "hyperautofluorescent crescent," "infinity sign," and "a broad banded hyper-autofluorescent leading edge." C: Singaporean patient, 51-year-old male with a BCVA of 0.4 (logMAR). Genetic variants c.6416G > A/c.5086G > T. Positive for "infinity sign" and "a broad banded hyper-autofluorescent leading edge." D: Singaporean patient, 60-year-old male with a BCVA of 0.0 (logMAR). Genetic variants c.6416G > A/c.904C > T. Displaying "hyperautofluorescent crescent," "infinity sign," and "a broad banded hyper-autofluorescent leading edge." Each image captures the distinct autofluorescence characteristics associated with *EYS*, providing insights into the phenotypic expression related to specific genetic mutations.

## Discussion

The present study revealed the distinct prevalence of specific FAF features in patients with *EYS*-associated RP. Notably, the so-called infinity sign and presence of a broad-banded hyper-autofluorescent leading edge were significantly more prevalent among *EYS*-associated RP patients. This finding highlights the potential of these biomarkers in the initial screening of RP for *EYS*-associated genetic contributions. Logistic regression analysis also showed that the infinity sign is an important indicator for assessing the probability of *EYS*-associated disease. Although the sensitivity was relatively low, the high specificity of these parameters suggests that a positive identification of features such as the infinity sign robustly indicates the presence of *EYS* variants in patients with RP. These results may allow for more targeted genetic investigations and refine diagnostic protocols in clinical settings, particularly in cases where a single *EYS* allele is identified by genetic testing and additional testing may be warranted to identify a second *EYS* allele to complete a genotype.

The infinity sign and broad-banded hyperAF leading edge emerged as potent indicators for *EYS*-associated RP. A previous report identified a crescent-shaped hyperautofluorescent leading edge and perifoveal hyperAF rings in *EYS*-associated RP [15], but our study suggests that these signs are less associated with *EYS* than the infinity sign or the broad-banded leading edge, at least in our East Asian study cohort. The larger sample size in our study may have enabled a more accurate analysis, reinforcing the value of these novel biomarkers for the identification of *EYS*-associated RP. It is worth pointing out that previous studies of European and North American *EYS* cohorts also revealed the presence of FAF features identified in our study, including the infinity sign and broad-banded leading edge, albeit without mention of these features as a specific sign of *EYS*-associated RP [15,17,18]. Moreover, previous work from Japan indicated that the presence of nasal sparing on FAF was relatively prevalent among RP cases among Japanese patients [19], which may directly reflect the high prevalence of *EYS*-associated RP in Japan. In that work, the presence of nasal sparing was suggested to be the result of cone photoreceptor enrichment in both the macula and the nasal peripapillary retina [20], which we likewise speculate is the underlying mechanism for the peculiar patterns of FAF seen in *EYS*-associated RP. Furthermore, recent work suggests that *EYS* may protect against phototoxicity in diurnal animals/humans [21]. We might speculate that the irregular AF patterns observed in *EYS*-associated RP are related in part to areas of more intensive retinal light exposure.

It should be noted that among the eight *PRPH2*-associated RP cases included in this study, the "infinity sign" was observed in three cases, and a broad-banded AF leading edge was observed in four cases, both of which occurred relatively frequently although it remains difficult to determine whether these FAF features are characteristic of *PRPH2*-associated RP because of its limited number. They share these characteristic FAF findings possibly because

both *EYS*, which is involved in the ciliary axoneme of photoreceptors and plays a pivotal role in the structural organization and maintenance of the outer segment (OS) [22], and *PRPH2*, which is essential for OS disc formation [23], both play important roles in preserving photoreceptor OS integrity. However, these two conditions have different inheritance patterns - EYS being Autosomal Recessive (AR) and PRPH2 typically being Autosomal Dominant (AD) [24]. In the presence of an AD pedigree, the FAF findings might point towards *PRPH2*, while in sporadic or AR cases, they would point more towards *EYS*. Therefore, FAF as a biomarker would still be useful for *EYS*.

Both the infinity sign and broad banded leading edge were found to be significantly associated with *EYS*-associated RP, although only the infinity sign was independently associated with *EYS* disease following logistic regression. We cannot exclude the possibility of an interaction between parameters with multicollinearity, wherein one parameter was judged to be insignificant because the effect of one parameter was nullified by another. FAF features may overlap, although it remains unclear what underlying pathophysiology is responsible for the clinical features observed here.

Our analysis confirmed that these specific FAF features effectively identified *EYS*-associated RP resulted with high specificity but low sensitivity and AUC. This shortcoming is important to note because these biomarkers markers will unlikely detect every case of *EYS* RP, but are highly specific when present. This robust specificity is especially valuable in clinical settings where accurate diagnosis can guide genetic counseling and tailor treatment strategies, potentially altering patient management. Furthermore, the establishment of an association between specific FAF features and specific *EYS* mutations could drive future research to explore automated image analysis techniques, further enhancing the efficiency and accuracy of diagnoses. Among the *EYS* variants observed in cases presenting with an "infinity sign" or a broad-banded hyperAF leading edge, c.2528G > A and c.4957dupA appeared relatively frequently, but these two variants are widely detected in general, and our current analysis did not detect a specific *EYS* variant significantly correlated with those FAF parameters.

Previous studies highlighted the utility of specific FAF patterns in diagnosing genetic subtypes of RP, associating characteristic autofluorescence features with specific genetic variants [14,25]. Conversely, *PRPF8* was reported to show variable expressivity of FAF features [26], resulting in overlapping phenotypes even within the same pathogenic gene, complicating the diagnostic process. This variability might also apply to other RP pathogenic genes, explaining the lower sensitivity observed in applying every FAF parameter in this study. We also hypothesize that the presence of more typical generalized RP among many of the *EYS* cases in the current study may be related to the presence of more pathogenic *EYS* variants, with milder variants unmasking the various FAF patterns we observed as biomarkers for *EYS* disease.

Our study had several limitations, including the difficulty in identifying FAF parameters consistently because the agreement of examiners graders were mostly "moderate" at best judged by kappa values, the relatively low sensitivity of the FAF markers, which limits their utility in early disease detection and may delay interventions for some patients. Our study included patients with early stage RP, late-stage RP, and cases with possible early cone dysfunction. Although FAF patterns may differ depending on disease stage, we did not perform stage-specific evaluations in this analysis. Additionally, the retrospective design may introduce biases related to case selection and data interpretation. These limitations suggest areas for improvement, such as prospective studies and advanced imaging technologies to enhance early detection and diagnostic accuracy. Furthermore, while the study was multinational with patients from Japan and Singapore, a larger study that includes patients from additional

geographic regions including Europe and the Americas would help to control for any genetic background effects that may influence the effect of *EYS* on FAF patterns in RP patients.

Despite these limitations, our focus on the correlation between specific FAF features and *EYS* gene mutations in RP is a significant strength. This specificity provides a foundation for more precise genetic and phenotypic profiling within this disease category. Additionally, the multicenter approach enhances the generalizability of the findings across different populations, demonstrating the relevance of this research in diverse clinical settings.

This study shows the diagnostic value of specific FAF features, notably the infinity sign and presence of a broad-banded leading edge, in identifying patients with *EYS*-associated RP. These features, while not universally present, appear to be highly specific and this is important in certain clinical settings, particularly in regions such as East Asia where *EYS*-associated RP is highly prevalent. Expanding this study to include more diverse patient populations and prospective designs may further validate and extend our findings. In conclusion, our study reveals the importance of specific autofluorescence markers in diagnosing *EYS*-associated RP, providing a useful tool for clinicians that may lead to improved patient outcomes through more precise and timely interventions in the future.

## Declaration of Generative AI and AI-assisted technologies in the writing process

During the preparation, this work used ChatGPT (GPT-4 September 25 version) in order to generate only a preliminary draft of the manuscript of this study. All subsequent drafts, data collection, citations, and interpretations were done by human researchers. After using this tool, the authors reviewed and edited the content and take full responsibility for the content of the publication.

## Supporting information

**S1 Fig. Distribution of *EYS* Variants in Japanese and Singaporean Populations.** The bar chart shows the distribution of specific *EYS* gene variants identified in Japanese and Singaporean patient cohorts. The black bars represent the number of occurrences of each pathogenic variants in the Japanese cohort, while the gray bars represent the Singaporean cohort. Each bar corresponds to the number of times a specific variant or combination of variants was observed in the each population.
(TIF)

**S2 Fig. Receiver operating curve (ROC) Curves for Predicting *EYS* vs. Non-*EYS* using Logistic Regression Models.** S2 Fig shows the Receiver operating curve (ROC) curves comparing two logistic regression models for distinguishing between patients with *EYS* mutations and those without. The Full ROC curve (black line) represents the model using both the baseline parameters (age at imaging, age at onset, sex, Va) and additional fundus autofluorescence (FAF) parameters (crescent, foveal, infinity, broadband, robson, sectoral). The Base ROC curve (gray line) represents the model using only the baseline parameters. The area under the curve (AUC) for the Full ROC curve is 0.53, indicating a improved discriminative ability compared to the Base ROC curve, which has an AUC of 0.42.
(TIF)

**S1 Table. *EYS* variants detected in cases with infinity sign and a broad-banded hyper AF leading edge.**
(TIF)

## Author contributions

**Conceptualization:** Taro Kominami, Koji M Nishiguchi, Beau J. Fenner.

**Data curation:** Taro Kominami, Tien-En Tan, Hiroaki Ushida, Kanika Jain, Kensuke Goto, Yasmin M. Bylstra, Ai Fujita Sajiki, Junya Ota, Weng Khong Lim, Beau J. Fenner.

**Formal analysis:** Taro Kominami, Tien-En Tan, Ai Fujita Sajiki, Beau J. Fenner.

**Funding acquisition:** Taro Kominami, Beau J. Fenner.

**Investigation:** Taro Kominami, Tien-En Tan, Hiroaki Ushida, Kanika Jain, Kensuke Goto, Yasmin M. Bylstra, Ranjana S. Mathur, Weng Khong Lim, Beau J. Fenner.

**Methodology:** Taro Kominami, Hiroaki Ushida, Yasmin M. Bylstra, Ai Fujita Sajiki, Ranjana S. Mathur, Beau J. Fenner.

**Project administration:** Taro Kominami, Beau J. Fenner.

**Resources:** Taro Kominami.

**Software:** Taro Kominami.

**Supervision:** Taro Kominami, Koji M Nishiguchi, Beau J. Fenner.

**Validation:** Taro Kominami.

**Visualization:** Taro Kominami.

**Writing – original draft:** Taro Kominami, Beau J. Fenner.

**Writing – review & editing:** Taro Kominami, Ai Fujita Sajiki, Ranjana S. Mathur, Junya Ota, Weng Khong Lim, Koji M Nishiguchi, Beau J. Fenner.

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
