## [Decision Letter · Decision Letter 0]

7 Jan 2025

PONE-D-24-48432Fundus autofluorescence features specific for EYS-associated retinitis pigmentosaPLOS ONE

Dear Dr. KOMINAMI,

Thank you for submitting your manuscript to PLOS ONE. After careful consideration, we feel that it has merit but does not fully meet PLOS ONE’s publication criteria as it currently stands. Therefore, we invite you to submit a revised version of the manuscript that addresses the points raised during the review process.

We look forward to receiving your revised manuscript.

Kind regards,

Tatsuya Inoue

Academic Editor

PLOS ONE

**Journal Requirements:**

This work was supported in part by the Japan Society for the Promotion of Science (JSPS)

KAKENHI, grant number 23K15929 to TK, the SingHealth Foundation (SHF) grant number R1748/71/2020 to BJF, and the SingHealth-Duke-NUS Institute of Precision Medicine grant numbers 05/FY2020/EX/06-A41 and 05/FY2022/EX(SLP)/69-A131(b) to KJ, YMB, and WKL. The funding organizations had no role in the design or conduct of this research.

"none"

4. In the online submission form, you indicated that your data is available only on request from a third party. Please note that your Data Availability Statement is currently missing the contact details for the third party, such as an email address or a link to where data requests can be made. Please update your statement with the missing information. 

Reviewers' comments:

Reviewer's Responses to Questions

**Comments to the Author**

1. Is the manuscript technically sound, and do the data support the conclusions?

Reviewer #1: Yes

Reviewer #2: Yes

2. Has the statistical analysis been performed appropriately and rigorously? 

Reviewer #1: Yes

Reviewer #2: Yes

3. Have the authors made all data underlying the findings in their manuscript fully available?

Reviewer #1: Yes

Reviewer #2: Yes

4. Is the manuscript presented in an intelligible fashion and written in standard English?

Reviewer #1: Yes

Reviewer #2: Yes

5. Review Comments to the Author

**Reviewer #1: ** The authors investigated the specific patterns of fundus autofluorescence achieved by an ultra-widefield ophthalmoscope in eyes with the eyes shut homolog (EYE)-associated retinitis pigmentosa (RP). They found out that the infinity sign, a specific infinity-shaped hyperautofluorescence, and the broad banded hyperautofluorescent leading edge pattern had high specificity (95.2%), while their sensitivity was not high. This research is important to screen out EYE-associated RP among RP patients. There may be some issues to be addressed. First of all, the infinity sign and the broad banded hyperautofluorescent leading edge predicted EYS only in 17 of 74 eyes (23.0%) and in 15 of 74 eyes (20.3%), respectively. On the other hand, non-EYS RP eyes also showed these autofluorescence patterns in 6 of 126 eyes and 8 of 126 eyes, respectively. Although these percentages were low as compared with those for EYS-associated RP, non-EYE eyes are sum of multiple genes. What kind of genes caused the non-EYS RP eyes with positive findings of these autofluorescence patterns? If some specific genes were related to these autofluorescence, the information might be fruitful to understand the relationship of pathogenesis and autofluorescence patterns.

(page 13, line 58) The abstract showed 100 patients with EYS and 100 without EYS, although the methods in the text stated 74 and 126, respectively.

(page 13, line 60) The comma after ‘the infinity sign’ should be ‘and’.

(page 18, line 126) The word, ‘resulting’ should be ‘resulting in’.

(page 21, line 169 and Table 1) The ratio of Singaporeans and Japanese should be shown in Table 1.

(page 21, line 173) The phrase in the parenthesis, ‘0.13 / 0.68…’ can be omitted because of statement in Table 1.

(Table 2) The total number is only 87. Were the pathogenic genes of the other 39 patients unknown? The authors should show all cases.

(Figure 2) P values for all groups should be shown in Figure 2.

(Table 4) What does ‘const’ mean?

(page 39) The reference, ‘Nakagawa S, …’ should be no. 19. Resultantly, the following no. 19-21 should be no. 20-22, respectively.

**Reviewer #2:**  Komami et al. compared the autofluorescence patterns in RP patients caused by EYS mutations with those in patients without EYS mutations. They found that the “infinity sign” was more frequently observed in RP cases caused by EYS mutations, suggesting its potential as a biomarker. Since EYS is the most frequent causative gene for RP in East Asia, identifying this characteristic is a significant discovery.

I have several comments to improve the manuscript. Although there are various variants in EYS gene, it would be helpful to show how these mutations are specifically associated with the infinity sign. Additionally, six cases of RP patients without EYS variants also exhibited the infinity sign. If possible, please identify the causative genes in these cases.

The autofluorescence patterns in RP would change during the disease progression. In this study, as noted in line 119, the cases include early-stage, late-stage, and cases with possible early cone dysfunction. However, the Discussion section does not address these changes. It is recommended to include a discussion of these changes.

Minor Points:

Line 250: Reference 19 is incorrect. Two references are included within Reference 18.

Please verify whether the numbering of subsequent references is correct.

6. PLOS authors have the option to publish the peer review history of their article (what does this mean? ). If published, this will include your full peer review and any attached files.

**Do you want your identity to be public for this peer review?** For information about this choice, including consent withdrawal, please see our Privacy Policy .

Reviewer #1: **Yes: ** Tsutomu Yasukawa

Reviewer #2: No

---

## [Author Response · Author response to Decision Letter 0]

17 Jan 2025

Reviewer #1:

Comment1

The infinity sign and the broad banded hyperautofluorescent leading edge predicted EYS only in 17 of 74 eyes (23.0%) and in 15 of 74 eyes (20.3%), respectively. On the other hand, non-EYS RP eyes also showed these autofluorescence patterns in 6 of 126 eyes and 8 of 126 eyes, respectively. Although these percentages were low as compared with those for EYS-associated RP, non-EYE eyes are sum of multiple genes. What kind of genes caused the non-EYS RP eyes with positive findings of these autofluorescence patterns? If some specific genes were related to these autofluorescence, the information might be fruitful to understand the relationship of pathogenesis and autofluorescence patterns.

Response1

Thank you for your insightful comment. Upon further review of our data, we found that both the “infinity sign” and a broad-banded hyperautofluorescent (AF) leading edge also appeared at a relatively high frequency in PRPH2-associated RP compared with other non-EYS RP although it remains challenging to conclude whether these FAF features are truly characteristic of PRPH2-associated RP owing to its limited number. They share characteristic FAF findings possibly because both EYS and PRPH2 are important for maintaining photoreceptor OS integrity. However, these two conditions have different inheritance patterns - EYS being Autosomal Recessive (AR) and PRPH2 typically being Autosomal Dominant (AD). In the presence of an AD pedigree, the FAF findings might point towards PRPH2, while in sporadic or AR cases, they would point more towards EYS. Therefore, FAF as a biomarker would still be useful for EYS. We have added sentences reflecting these considerations to the Results (page 18 and 19, lines 200–204) and Discussion (pages 25 and 26, lines 301–313).

Comment2

(page 13, line 58) The abstract showed 100 patients with EYS and 100 without EYS, although the methods in the text stated 74 and 126, respectively.

Response2

Thank you for pointing this out to us. We have corrected the number of EYS and non-EYS (page5, lines 52 and 53).

Comment3

(page 13, line 60) The comma after ‘the infinity sign’ should be ‘and’.

Response3

Thank you for your comment. We have corrected this (page5, line 54).

Comment4 (page 18, line 126) The word, ‘resulting’ should be ‘resulting in’.

Response4

Thank you for your comment. We have corrected this (page10, line 120).

Comment5 (page 21, line 169 and Table 1) The ratio of Singaporeans and Japanese should be shown in Table 1.

Response5

Thank you for your comment. We have added the ratio of Singaporeans to Japanese to Table 1.

Comment6 (page 21, line 173) The phrase in the parenthesis, ‘0.13 / 0.68…’ can be omitted because of statement in Table 1.

Response6

Thank you for your comment. We have omitted this (page14, line 178).

Comment7 (Table 2) The total number is only 87. Were the pathogenic genes of the other 39 patients unknown? The authors should show all cases.

Response7

Thank you for pointing this out. We apologize for the confusion. If you simply add up the “number” column on the right side of the table, as you mentioned, it totals 87. However, for example, in the fourth row from the top, both RHO and RPGR are listed as seven, meaning that each gene accounts for seven cases, resulting in a total of 14. In other rows where multiple genes are listed, the number of cases for each gene is also reported separately. Consequently, if you consider the sum of each gene type on the left and the corresponding number on the right, the total number is 126. We have revised the term “number” in Table 2 to “Number of each gene” to make it clearer.

Comment8 (Figure 2) P values for all groups should be shown in Figure 2.

Response8

Thank you for your comment. We have added P-values to Figure2.

Comment9 (Table 4) What does ‘const’ mean?

Response9

Thank you for your comment. The “const” is a constant term obtained from the logistic regression results and does not hold any significance as an FAF parameter, so we have omitted it from Table 4.

Comment10 (page 39) The reference, ‘Nakagawa S, …’ should be no. 19. Resultantly, the following no. 19-21 should be no. 20-22, respectively.

Response10

Thank you for pointing this out to us. We apologize for the errors in the reference numbers and list. We have corrected them accordingly(page27, lines 336 and 337; page34-35, lines 444-463). 

Reviewer #2:

Comment1

Although there are various variants in EYS gene, it would be helpful to show how these mutations are specifically associated with the infinity sign.

Response1

Thank you for pointing this out to us. We summarized variants of EYS cases with “infinity sign” and a broad-banded hyperautofluorescent (AF) leading edge in the supplemental table. Among the EYS variants observed in cases presenting with an “infinity sign” or a broad-banded hyperAF leading edge, c.2528G>A and c.4957dupA appeared relatively frequently. However, these two variants are widely detected in general, and our current analysis did not detect a variant that was significantly correlated with FAF parameters. We have added sentences to the Results section(page18-19, lines 204-205) and Discussion section(page27, lines 329-333).

Comment2

Additionally, six cases of RP patients without EYS variants also exhibited the infinity sign. If possible, please identify the causative genes in these cases.

Response2

Thank you for your comments. Upon further review of our data, we found that the “infinity sign” also appeared at a relatively high frequency in PRPH2-associated RP compared with other non-EYS RP although it remains challenging to conclude whether “infinity sign” is truly characteristic of PRPH2-associated RP owing to its limited number. They share characteristic FAF findings possibly because both EYS and PRPH2 are important for maintaining photoreceptor OS integrity. However, these two conditions have different inheritance patterns - EYS being Autosomal Recessive (AR) and PRPH2 typically being Autosomal Dominant (AD). In the presence of an AD pedigree, “infinity sign” might point towards PRPH2, while in sporadic or AR cases, “infinity sign” would point more towards EYS. Therefore, FAF as a biomarker would still be useful for EYS. We have added sentences reflecting these considerations to the Results (page 18 and 19, lines 200–204) and Discussion (pages 25 and 26, lines 301–313).

Comment3

The autofluorescence patterns in RP would change during the disease progression. In this study, as noted in line 119, the cases include early-stage, late-stage, and cases with possible early cone dysfunction. However, the Discussion section does not address these changes. It is recommended to include a discussion of these changes.

Response3

Thank you for your comment. We have added the sentences to address this point in the Discussion section (page28, lines348-350).

Comment4

Line 250: Reference 19 is incorrect. Two references are included within Reference 18.Please verify whether the numbering of subsequent references is correct.

Response4

Thank you for pointing this out to us. We apologize for the errors in the reference numbers and list. We have corrected them accordingly(page27, lines 336 and 337; page34-35, lines 444-463).

---

## [Decision Letter · Decision Letter 1]

23 Jan 2025

Fundus autofluorescence features specific for EYS-associated retinitis pigmentosa

PONE-D-24-48432R1

Dear Dr. KOMINAMI,

We’re pleased to inform you that your manuscript has been judged scientifically suitable for publication and will be formally accepted for publication once it meets all outstanding technical requirements.

Kind regards,

Tatsuya Inoue

Academic Editor

PLOS ONE

Additional Editor Comments (optional):

Reviewers' comments:

Reviewer's Responses to Questions

**Comments to the Author**

1. If the authors have adequately addressed your comments raised in a previous round of review and you feel that this manuscript is now acceptable for publication, you may indicate that here to bypass the “Comments to the Author” section, enter your conflict of interest statement in the “Confidential to Editor” section, and submit your "Accept" recommendation.

Reviewer #1: All comments have been addressed

Reviewer #2: All comments have been addressed

2. Is the manuscript technically sound, and do the data support the conclusions?

Reviewer #1: Yes

Reviewer #2: Yes

3. Has the statistical analysis been performed appropriately and rigorously? 

Reviewer #1: Yes

Reviewer #2: Yes

4. Have the authors made all data underlying the findings in their manuscript fully available?

Reviewer #1: Yes

Reviewer #2: Yes

5. Is the manuscript presented in an intelligible fashion and written in standard English?

Reviewer #1: Yes

Reviewer #2: Yes

6. Review Comments to the Author

Reviewer #1: (No Response)

Reviewer #2: The authors responded to my comment properly. The contents of this seems to be very interesting. I have no other comments.

7. PLOS authors have the option to publish the peer review history of their article (what does this mean? ). If published, this will include your full peer review and any attached files.

**Do you want your identity to be public for this peer review?** For information about this choice, including consent withdrawal, please see our Privacy Policy .

Reviewer #1: **Yes: ** Tsutomu Yasukawa

Reviewer #2: No

---

## [Editor Report · Acceptance letter]

PONE-D-24-48432R1

PLOS ONE

Dear Dr. Kominami,

I'm pleased to inform you that your manuscript has been deemed suitable for publication in PLOS ONE. Congratulations! Your manuscript is now being handed over to our production team.

Kind regards,

on behalf of

Dr. Tatsuya Inoue

Academic Editor

PLOS ONE